# Photodegradation of Riboflavin under Alkaline Conditions: What Can Gas-Phase Photolysis Tell Us about What Happens in Solution?

**DOI:** 10.3390/molecules26196009

**Published:** 2021-10-03

**Authors:** Natalie G. K. Wong, Chris Rhodes, Caroline E. H. Dessent

**Affiliations:** Department of Chemistry, University of York, Heslington YO10 5DD, UK; natalie.wong@york.ac.uk (N.G.K.W.); chris.rhodes@york.ac.uk (C.R.)

**Keywords:** laser spectroscopy, on-line photolysis, solution-phase photolysis, vitamin, flavins, mass spectrometry

## Abstract

The application of electrospray ionisation mass spectrometry (ESI-MS) as a direct method for detecting reactive intermediates is a technique of developing importance in the routine monitoring of solution-phase reaction pathways. Here, we utilise a novel on-line photolysis ESI-MS approach to detect the photoproducts of riboflavin in aqueous solution under mildly alkaline conditions. Riboflavin is a constituent of many food products, so its breakdown processes are of wide interest. Our on-line photolysis setup allows for solution-phase photolysis to occur within a syringe using UVA LEDs, immediately prior to being introduced into the mass spectrometer via ESI. Gas-phase photofragmentation studies via laser-interfaced mass spectrometry of deprotonated riboflavin, [RF − H]^−^, the dominant solution-phase species under the conditions of our study, are presented alongside the solution-phase photolysis. The results obtained illustrate the extent to which gas-phase photolysis methods can inform our understanding of the corresponding solution-phase photochemistry. We determine that the solution-phase photofragmentation observed for [RF − H]^−^ closely mirrors the gas-phase photochemistry, with the dominant *m*/*z* 241 condensed-phase photoproduct also being observed in gas-phase photodissociation. Further gas-phase photoproducts are observed at *m*/*z* 255, 212, and 145. The value of exploring both the gas- and solution-phase photochemistry to characterise photochemical reactions is discussed.

## 1. Introduction

Riboflavin (RF; Figure 1) is one of the most well-studied members of the flavin family and is a vital water-soluble vitamin (B_2_) naturally found in a wide range of food products (e.g., milk, cheese, and green vegetables) and fortified foods (e.g., cereals, bread, and beer). As a precursor of all biologically-significant flavins, riboflavin is an integral component of the (flavo)coenzymes flavin adenine dinucleotide (FAD) and flavin mononucleotide (FMN), both of which are active within several critical metabolic enzyme reactions.

Riboflavin shows broad absorption across the UV–visible regions in aqueous solution and has been observed to photodegrade via intramolecular and/or intermolecular photoreduction, photoaddition, and photodealkylation mechanisms to form an array of photoproducts including formylmethylflavin (FMF), lumichrome (LC), lumiflavin (LF), carboxymethylflavin (CMF), cyclodehydroriboflavin (CDRF), and 2,3-butanedione, as reviewed in detail elsewhere [1]. These studies have revealed that the formation of intermediates is heavily dependent on reaction conditions (i.e., pH, solvent, and light intensity), and that the intermediates have the capacity to degrade into secondary products [1,2].

In the body, riboflavin has an absorption maximum of ca. 447 nm. Human skin, which contains significant levels of riboflavin, can thereby photodegrade upon exposure to blue (visible) light, causing red-cell lysis and generalised tissue riboflavin deficiencies within the body. Earlier work on riboflavin has shown that it can act as an avid photosensitiser, yielding reactive oxygen species (ROS) such as the superoxide anion, hydroxy radical, hydrogen peroxide, as well as singlet oxygen. Such radicals can in turn promote the decomposition of other ingredients within food, pharmaceuticals, and cosmetics. Recent work investigating the influence of UV radiation on the RF photochemistry has shown that riboflavin directly produces singlet oxygen at 308, 330, 355, and 370 nm [3], explaining why riboflavin levels in food can rapidly degrade upon exposure to both natural and artificial light, thus limiting its shelf-life. Given its prominence within many metabolic processes, it is of ongoing interest to better understand the photochemical breakdown pathways of riboflavin in order to develop better ways to stabilise the vitamin in vivo and within consumer products.

The condensed-phase photostability of riboflavin at various pH values (i.e., (de)protonation state) has been investigated previously: under acidic/neutral conditions riboflavin photodegrades to LC whereas, under alkaline conditions, it photodissociates to both LC and LF. Whilst LC is formed via a riboflavin excited singlet state (^1^RF *), both LF and LC can be formed via intramolecular photoreduction from an excited triplet state (^3^RF *). The pK_a_ values of riboflavin are 1.7 and 10.2, and its breakdown kinetics have been studied extensively across the corresponding pH ranges, showing that degradation increases by ca. 80 fold in alkaline media (correlated with the increased reactivity of the triplet state) until pH 10, when the anion is formed [4].

Unsurprisingly, the extensive solution-phase photochemical behaviour of riboflavin has accrued much experimental attention over the years [1,5]; however, studies of its intrinsic (i.e., gas-phase) excited-state properties, and those of other flavins, have been sparse until very recently [6,7,8,9,10,11,12,13]. To date, the only gas-phase study of a system involving riboflavin is that of Dopfer and co-workers on alkali metal coordination [9]. This situation is significant given the need for gas-phase studies as benchmarks for computational work, which can provide a broader fundamental understanding of RF photophysics. For anionic flavin systems, Matthews et al. have compared the intrinsic electronic behaviour of deprotonated LC and its related chromophore alloxazine (AL), finding novel near-threshold transient anion resonance states, which were assigned to dipole-bound excited states [6]. Recently, Stockett and co-workers have utilised tandem ion mobility spectrometry and action spectroscopy to probe the influence that deprotonation sites have on the electronic absorption properties and photochemistry of the FAD anion [8], following on from earlier experimental work assigning probable sites of (de)protonation of flavins in the gas phase [9,10,11,12,13,14]. Non-statistical (i.e., occurring during the excited-state lifetime) LC formation of the FAD mono-anion was found to proceed through a photo-induced intramolecular proton-coupled electron transfer of riboflavin [9].

In this article, we present the first UV–visible electronic gas-phase study of the deprotonated form of riboflavin. We employ the technique of gas-phase laser photodissociation action spectroscopy on [RF − H]^−^ through forming the anion as an isolated gas-phase species via electrospray ionisation (ESI), and then obtaining the electronic absorption spectrum and action spectra of its resultant photofragment ions. Our gas-phase measurements are complemented by on-line photolysis in a UV-LED photolytic cell (365 nm) with ESI mass spectrometry (ESI-MS) [15]. This experimental approach, which characterises the gas- and solution-phase behaviour of a chosen system, provides a “one-pot” tool for approaching the understanding of photochemistry. The technique builds on other recent experiments which have employed on-line photolysis, with subsequent mass spectrometric detection [16,17,18,19,20]. The benefit of this approach is that the solution-phase measurements probe a more “real-world” environment for the riboflavin vitamin (e.g., when it is in milk or within a cosmetic formulation), while the gas-phase measurements aid the understanding of the solution-phase photochemical photoproducts. The gas-phase photodissociation methodology employed here has an advantage compared to solution-phase measurements, as it allows the species of interest to be isolated prior to interacting with the laser. This ensures that photofragments observed are produced from the species of interest, and not from a minor component of the solution-phase mixture. Furthermore, the solution environment is potentially much more complex, since direct photoproducts can undergo secondary reactions and will be subject to environmental factors, i.e., pH, solvent, and aggregation [5,16,17,18,19,20]. Herein, we evaluate the benefits of this two-step methodology for studying photochemistry by applying it to RF in its native solution-phase form, [RF − H]^−^ We have chosen RF for this study since it’s solution photochemistry has been well studied, thus allowing us to benchmark our solution-phase on-line photolysis technique and also understand if the gas-phase measurements can shed light on the solution-phase photochemistry. The two-step photolysis method provides a new approach that could be widely applied to characterise the photochemical breakdown of food, beauty, and pharmaceutical products. 

## 2. Results

### 2.1. Gas-Phase Absorption Spectroscopy of [RF − H]^−^

Figure 1 shows the electrospray ionisation mass spectrum of a solution of riboflavin obtained in the negative ion mode. [RF − H]^−^ appears as the dominant ion in the mass spectrum, with *m*/*z* 375 (Figure 1), along with a second intense fragment ion with *m*/*z* 255, which can be assigned as deprotonated lumiflavin, [LF − H]^−^. (Intriguingly, its corresponding protonated species is not observed at all in the positive ion mode; Appendix A). The dimer complex, i.e., [RF − H]^−^ RF, is observed at *m*/*z* 751.

Laser-interfaced mass spectrometry (LIMS) was used to characterise the wavelength-dependent photoabsorption and photofragmentation properties of [RF − H]^−^ in vacuo. The spectra are acquired by action spectroscopy, as described in Section 4 and in References [21,22,23]. Figure 2a displays the gas-phase photodepletion spectrum of [RF − H]^−^, which can be considered to be the gas-phase UV–visible absorption spectrum of the ion, subject to the limitations discussed in References [22,24,25,26]. The absorption spectrum displays an absorption onset at 2.0 eV (620 nm), followed by five bands labelled **I**–**V**. Bands **I**–**III** (at ca. 3.0, 3.5, and 4.2 eV, respectively) are medium-intensity, broad bands, with band **I** observed in the visible region, centred at approximately 3.0 eV (414 nm). Absorption continues to extend across into the UVC region, where band **IV** appears as a relatively strong feature between 4.5 and 5.2 eV, peaking at 5.0 eV (246 nm). The rising edge of band **V** is at the high-energy spectral edge and extends beyond 5.74 eV (216 nm).

The absorption profile of [RF − H]^−^ in aqueous solution is shown in Figure 2b, for comparison with the gaseous spectrum (Figure 2a). RF has multiple acidic protons, and its reported pK_a_ values of 1.7 and 10.2 [27] indicate that it will be deprotonated in aqueous solution at neutral pH. Drӧssler et al. have demonstrated that the solution-phase UV absorption spectra of RF in aqueous solution is unchanged between the pH values of 7.0, 9.3, 10.25, and 13.35 [28]. 

On comparing the experimental gaseous and solution-phase spectra in Figure 2a,b, respectively, both spectra are seen to display relatively low absorption across the visible-UVB regions, with the two dominant gas-phase UV bands (**IV** and **V**) likely to correspond to the strong **III** and **IV** features seen in the solution phase. It is striking that the intense solution-phase transition band **III** seems to appear much less strongly in the gas-phase spectrum (4.3 eV). This effect can arise from fluorescence decay of the electronic states associated bands which are not observed in the gas phase when action spectroscopy is used to acquire the absorption spectrum [26]. Indeed, riboflavin is known to exhibit fluorescence, with its anionic form in particular, found to exhibit a fluorescence quantum yield of 1.2 × 10^−3^ in aqueous solution [28].

### 2.2. Gas-Phase Photofragmentation of [RF − H]^−^

We next turn to exploring the gas-phase photofragment ions that are associated with the excited states evident in the gaseous absorption spectrum (Figure 2a). Figure 3a–e display the photofragment mass spectra of [RF − H]^−^ obtained following excitation at 2.5, 3.5, 4.2, 4.7, and 5.0 eV. The spectra show that ions at *m*/*z* 255 and 241 are observed as the dominant and minor ionic fragments, respectively, corresponding to a loss of 120 and 134 Da from [RF − H]^−^ to form deprotonated lumiflavin ([LF − H]^−^) and lumichrome ([LC − H]^−^), respectively. It has been previously noted that both LF and LC are formed in alkaline media via the triplet excited state via formylmethylflavin (FMF; 284.27 Da), which is believed to be an intermediate in the photolysis of RF. Several less intense, minor fragment ions are also observed, e.g., at *m*/*z* 212 and 145.

The major photodissociation channels are summarised in Equations (1)–(4) and again in Table 1.
[RF − H]^−^ + h*v*→*m*/*z* 255 + C_4_H_8_O_4_(1)
*m*/*z* 241 + C_5_H_10_O_4_
(2)
*m*/*z* 212 + C_4_H_8_O_4_ + HNCO (3)
*m*/*z* 145 + C_8_H_10_N_2_O_6_
(4)

Since [RF − H]^−^ is anionic, any photodepletion that is not associated with photofragment ion production must generally be associated with electron loss processes [6,29,30]. This is discussed further in Appendix A. 

Our LIMS setup allows us to monitor the photofragment production intensities at each scanned wavelength to provide further insight into the nature of the excited states. Figure 4b displays the production spectrum of the major photofragment [LF − H]^−^ with *m*/*z* 255, showing that it is produced across the entire experimental spectral range from 2.0–5.74 eV (620–216 nm), with significant peaks in production at the band maxima **I**–**V** of the gas-phase absorption spectrum of [RF − H]^−^ (Figure 4a for ease of comparison). The photofragment production spectra of the minor fragments at *m*/*z* 241 and 212 (Figure 4c,d, respectively) also display similar absorption maxima at 3.5, 5.0, and 5.7 eV, but with distinctive absorption profiles in the lower-energy region ca. 2.5 eV compared to the *m*/*z* 255 ion. Little to no production of these fragments occurs between 4.0 and 4.5 eV. The production spectrum of the *m*/*z* 145 photofragment (Figure 4e) has a production onset at ~2.75 eV, subsequently peaking at 3.25, 5.0 and 5.7 eV.

Figure 5 presents the relative photofragment ion yields for [RF − H]^−^ as a function of excitation energy, revealing several maxima that can be attributed to photoexcitation into different electronic states [31,32]. It is evident that between 2.0 and 3.0 eV, the *m*/*z* 241 and 212 photofragment ions are dominant, whereas the *m*/*z* 145 ion is not produced across this region. Between 3.0 and 4.0 eV, all three fragments are produced, with *m*/*z* 212 produced with ~50% greater intensity than that of the *m*/*z* 145 ion. In contrast, between 4.0 and 5.74 eV, *m*/*z* 145 overtakes both the *m*/*z* 241 and 212 ions in relative ion intensity, with all three increasing steadily with photon energy towards the high-energy edge.

### 2.3. Collision-Induced Dissociation of [RF − H]^−^

Low-energy collision-induced dissociation (CID) of [RF − H]^−^ gives [LF − H]^−^ (*m*/*z* 255) as the only ionic product (Figure 6). The CID process is equivalent to thermal heating of ground state [RF − H]^−^, with subsequent fragmentation across the lowest energy fragmentation barrier on the ground state surface [33]. The precursor [RF − H]^−^ ion does not dissociate up to ~40% CID, indicating that the ion is stable against spontaneous dissociation. However, as noted in Section 2.1, ESI-MS of riboflavin produces *m*/*z* 255 as well as [RF − H]^−^, and the observation of *m*/*z* 255 as a CID fragment indicates that [RF − H]^−^ is likely fragmenting into *m*/*z* 255 either during ESI or via in-source dissociation. It is also notable that *m*/*z* 255 is the major gas-phase laser photofragment ion.

To further probe the thermal fragmentation pathway (s) of [RF − H]^−^ on its electronic ground state, higher-energy collisional dissociation (HCD) was employed. These measurements are ideal to identify which ions are secondary fragments, formed when a precursor (or intermediate) species fragments at high internal energy. Importantly, any photofragments not observed in these HCD experiments can be identified as purely photochemical products. A HCD plot of [RF − H]^−^ electrosprayed in EtOH in the negative ion mode is shown in Figure 7, revealing the nature of the thermal degradation products at higher HCD energies. (We note that it was not possible to electrospray from an aqueous solution in these experiments, due to differences in the ionisation efficiencies of the two different instruments used. However, it was possible to electrospray from both EtOH and MeCN in the HCD instrument. The results obtained with both solvents were extremely similar, giving us confidence that this would also be the case if the electrospray had been conducted from an aqueous solution.)

At relatively low collisional energies (0−25% HCD), the most intense fragment ion is *m*/*z* 255, with *m*/*z* 243 peaking at ~15% higher relative ion intensity. Production of the *m*/*z* 255 ion gradually decreases beyond 25% HCD energy, where the onset of the *m*/*z* 212 ion occurs. This secondary ion subsequently peaks at 48% HCD energy. However, since the *m*/*z* 255 fragment clearly dominates in both the gas-phase laser and thermal (i.e., CID and HCD) fragmentation experiments, photofragmentation of [RF − H]^−^ can be categorised as predominantly statistical (ergodic) over the spectral range of this experiment [31,34].

Overall, the CID and HCD fragmentation experiments reveal that [RF − H]^−^ primarily dissociates into the *m*/*z* 255 ion, which subsequently dissociates into *m*/*z* 212 at higher internal energy, and increasingly into *m*/*z* 241 and 145 at the highest internal energies. 

### 2.4. On-Line Solution-Phase Photolysis of [RF − H]^−^

Figure 8 shows photolysis_on_—photolysis_off_ (difference) ESI mass spectrum obtained from the photolysis (UV-A; 365 nm) of aqueous riboflavin at 3, 6, 9, and 12 min. The [RF − H]^−^ precursor ion (*m*/*z* 375), the primary thermal ion (*m*/*z* 255), and the dimer ion (*m*/*z* 751) are all seen to be depleted in the difference spectra, indicating that they are photolyzed at this wavelength. Solution-phase photolysis of [RF − H]^−^ predominantly produces the same *m*/*z* 241 ion that was observed in the gas-phase irradiation experiments, which corresponds to [LC − H]^−^. The ion at *m*/*z* 617 is again the loss of C_5_H_10_O_4_ (134 Da) from the dimer complex [RF − H]^−^ RF at *m*/*z* 751. The same loss in *m*/*z* is observed in Equation (2).

Figure 9 shows the relative ion intensities of the solution-phase photofragment ions observed through ESI-MS as a function of irradiation time. The relative intensities of the ions which are initially present upon electrospray (i.e., *m*/*z* 255, 375, and 751) remained steady until ca. 3.0 min (the transit time of the solution that is being photolyzed into the MS), where the sudden production onset of the major *m*/*z* 241 fragment ion is then observed. Several relatively minor fragments (<5% relative ion intensity over 30 min of irradiation) are also observed, as outlined in Appendix A.

Control experiments run in parallel using routine ESI-MS detection showed that under the same photolytic conditions, solutions of [RF − H]^−^ held within “black-coated” syringes (e.g., to block all transmission of UV light) showed no degradation of the parent (*m*/*z* 375) ion, nor the *m*/*z* 255, or *m*/*z* 375 ions (see Appendix A). The solution-phase photofragment at *m*/*z* 241 identified earlier in Figure 8 and Figure 9 was also not observed. This demonstrated that these ions remained unaffected by any other potential external degradation pathways (e.g., heating or hydrolysis). 

## 3. Discussion

In this work, gas-phase laser-interfaced photodissociation mass spectrometry has been employed on the native form of riboflavin for the first time to map its intrinsic UV–visible absorption (photodepletion) spectrum and associated photodegradation products. Solution-phase photoproducts were generated via photolysis on-line with the MS instrument, and were characterised using ESI-MS and compared with the gas-phase results. The motivation for this work was to benchmark our on-line photolysis apparatus for a system where the solution-phase photochemistry is well known (i.e., RF), and then to compare the solution-phase and gas-phase photofragmentation to understand the benefit of conducting both measurements together.

The on-line, solution-phase photolysis results we obtained clearly match previous solution-phase experiments conducted for RF, demonstrating that our on-line photolysis apparatus can reliably be applied to probe solution-phase photoproducts. In comparing these solution-phase results to our gas-phase results, we found that whilst [RF − H]^−^ photodegrades into deprotonated lumichrome (*m*/*z* 241) via the loss of 134 Da in both the gas and solution phases at 365 nm (3.4 eV), further ions at *m*/*z* 255, 212, and 145 were observed to be produced following gas-phase photodissociation of the isolated molecular ion. (Our HCD results revealed that the *m*/*z* 255 ion dissociates into *m*/*z* 212 and 145 at high internal energies.) Importantly, we have observed that the major gas-phase photofragment, *m*/*z* 255, deprotonated lumiflavin, is not produced to any significant effect in solution under alkaline conditions. This reveals that an energy dissipation pathway exists in solution to protect RF from breaking down into the 255 *m*/*z* product. 

The photoproducts observed in this work mirror those seen in related work by Insińska-Rak et al. [17,20], where pH neutral solutions of riboflavin and its derivatives were photolyzed by mercury lamps, and photoproducts detected by HPLC and electron ionisation mass spectrometry (EI-MS). The fragmentation pathways of riboflavin proceed with via formation of lumichrome and lumiflavin, with major ions observed at *m*/*z* 256 (relative intensity of 33%), 213 (43%), 198 (52%), and 170 (25%) via the lumiflavin route, or *m*/*z* 242 (12%), 199 (28%), 171 (42%), and 156 (16%) through the lumichrome route [17,20]. (Note that such mass-to-charge ratios are monitored via EI-MS). All of the photofragments observed in the work of Insińska-Rak et al. are in line with the main photoproducts formed in this work (with the exception of our *m*/*z* 145 photofragment, which we know is a secondary fragment).

The results presented herein therefore demonstrate an elegant new technique for monitoring solution-phase photodegradation of [RF − H]^−^ and detection of its resulting ionic photodissociation products. The direct and uninterrupted injection of the irradiated contents of the syringe allows for continuous real-time monitoring of the change in ion intensities of select ions over time (see Figure 9). This work also demonstrates the value of performing the gas-phase photodissociation measurements, even in the absence of on-line solution-phase photolysis, since the gas-phase measurement allows the identification of the major photolysis product for this system, and hence potentially others. The gas-phase measurements have several advantages as they are conducted against zero background, hence allowing straightforward identification of minor photoproducts, as well as allowing the primary photoproducts to be identified away from secondary reactions that can occur in solution. Nonetheless, the combination of on-line solution-phase photolysis coupled with gas-phase photolysis of the mass-selected species of interest clearly allows a rapid and straightforward link to be established between the gas-phase measurement and the “real world” environment of the solution photochemistry. This approach has potential for widespread application across photochemical systems of interest, including photopharmaceuticals [35] and sunscreens [23,36,37].

## 4. Materials and Methods

**Laser-Interfaced Photodissociation Mass Spectrometry.** The gaseous ion absorption (photodepletion) and photofragment production spectra of [RF − H]^−^ were recorded in vacuo using laser action spectroscopy. An AmaZon SL mass spectrometer (Bruker Daltonics Inc., Billerica, MA, USA), modified for laser-interfaced mass spectrometry (LIMS), was used as described previously [22]. RF was purchased from Sigma-Aldrich (St. Louis, MO, USA) and used as received. RF solutions (1 × 10^−5^ M) in deionised H_2_O were electrosprayed at a capillary temperature 100 °C in the negative ion mode. Trace amounts of NH_3_ (0.4%) was added to the aqueous solution (s) to aid the electrospray process, as the intensity of ions produced without the additive deemed insufficient for later gas-phase measurements.

[RF − H]^−^ was mass-selected (*m*/*z* 375) and isolated in the ion trap prior to laser irradiation. Photons were produced by an Nd: YAG pumped OPO laser (Surelite™/Horizon™, Amplitude Laser Group, San Jose, CA, USA), giving 0.1 ± 15% mJ across the UV–visible range of 620–216 nm (2.0–5.74 eV), with 2 nm laser step sizes in the UV and 4 nm steps in the visible. Photofragmentation experiments were conducted with an ion accumulation time of 20 ms. To minimise the possibility of multiphoton events via sequential absorption, each mass-selected ion packet interacts with only one laser pulse (corresponding to a fragmentation time of 100 ms), and photodepletion restricted to ∼40 % of the precursor ion at the wavelength of maximum absorption. Multiphoton events via instantaneous absorption of multiple photons in the Franck–Condon region are negligible as the laser beam is only softly focused through the ion-trap region.

Photodepletion of [RF − H]^−^ was measured as a function of the scanned wavelength, with photofragment production recorded simultaneously Equations (5) and (6):(5)$$\mathrm{Photodepletion}\;(\mathrm{PD})\;\mathrm{Intensity}=\frac{\ln(\frac{{\mathrm{Int}}_{\mathrm{OFF}}}{{\mathrm{Int}}_{\mathrm{ON}}})}{\lambda \;\times \;P}$$
(6)$$\mathrm{Photofragment}\;(\mathrm{PF})\;\mathrm{Production}\;\mathrm{Intensity}=\frac{\left(\frac{{\mathrm{Int}}_{\mathrm{FRAG}}}{{\mathrm{Int}}_{\mathrm{OFF}}}\right)}{\lambda \;\times \;P}$$

In these expressions, Int_OFF_ and Int_ON_ are the laser off and on parent ion peak intensities, respectively; Int_FRAG_ is the fragment intensity with the laser on; λ is the excitation wavelength (nm); P is the laser pulse energy (mJ); and Int_PFT_ is the sum of the photofragment ion intensities with the laser on. The photodepletion spectrum is considered to be equivalent to the gaseous absorption spectrum in the limit where excited-state fluorescence is negligible.

Electron detachment yield (ED*) spectra were calculated by assuming that any depleted ions not detected as ionic photofragments are decaying via means of electron detachment, as determined using Equation (8). This analysis assumes that both the parent ions and photofragments are detected equally in the mass spectrometer. In Appendix A, where we present ED* spectra, we overlay such data with the photodepletion yield (PD*). PD* is the normalised photodepletion ion count Equation (9), which provides the most straightforward comparison to the electron detachment yield Equation (8).
(7)$$\mathrm{Electron}\;\mathrm{detachment}\;\mathrm{yield}\;(\mathrm{ED}*)=\frac{\frac{({\mathrm{Int}}_{\mathrm{OFF}}{\;-\;\mathrm{Int}}_{\mathrm{ON}}{)\;-\;\mathrm{Int}}_{\mathrm{PFT}}}{{\mathrm{Int}}_{\mathrm{OFF}}}\;}{\lambda \;\times \;P}$$
(8)$$\mathrm{Photodepletion}\;\mathrm{yield}\;(\mathrm{PD}*)=\frac{\frac{{\mathrm{Int}}_{\mathrm{OFF}}{\;-\;\mathrm{Int}}_{\mathrm{ON}}}{{\mathrm{Int}}_{\mathrm{OFF}}}}{\lambda \;\times \;P}$$

**Analysis.** Photodepletion intensities were taken from an average of three repeat runs at each wavelength of the range studied. We note that fragment ions with *m*/*z* < 50 are not detectable in our mass spectrometer since low masses fall outside the ion-trap mass window.

**Collision-Induced Dissociation Energy.** Low-energy collision-induced dissociation (CID) was performed to investigate the ground-state thermal fragmentation characteristics of [RF − H]^−^, using the aforementioned Bruker AmaZon mass spectrometer. Here, an excitation AC folder was applied to the end caps of the ion trap to induce low-energy collisions of the trapped anions with the helium buffer gas, as has been described in detail previously [33,38]. Precursor ion excitation within the quadrupole ion trap occurs through multiple low-energy collisions with helium. Resonance excitation amplitudes for thermal decomposition of molecular ions within a quadrupole ion trap have been shown to correlate with literature critical dissociation energies. However, given that is no direct conversion from the resonance excitation voltage to absolute CID energies, such CID energies are therefore quoted as a percentage of the 2.5 V excitation voltage, in line with other standard practices [33,38,39,40].

**Higher-energy Collisional Dissociation.** Higher-energy collisional dissociation (HCD) was performed on [RF − H]^−^, electrosprayed from solutions of EtOH and MeCN using an Orbitrap Fusion Tribid mass spectrometer (Thermo Fisher Scientific, Waltham, MA, USA) with an ESI source, run in the negative ion mode between 0% and 60% collisional energy. This technique provides tandem mass spectrometry and was operated at a flow rate of 3.0 µL/min, with the following parameters: spray voltage: −2000 V; sheath gas flow rate: 10; aux. gas flow rate: 2.0; ion transfer tube temperature: 290 °C; vaporiser temperature: 20 °C; MS^2^ detector: Orbitrap; scan rate: enhanced; MS^2^ injection time: 100 ms; and RF lens: 60%.

**Solution-Phase UV-A Photodissociation.** For sample irradiation, four UVA light-emitting diodes (LEDs; 365 nm {3.4 eV}, LZ1–00UV00, LED Engin, Inc., San Jose, CA, USA) were employed. The four LEDs were connected in series and powered with a 500 mA constant current power source. Each of the LEDs were situated onto each face of the square (23 mm × 23 mm) interior of the custom designed aluminium cell, surrounding the barrel of the UV-transmitting borosilicate glass of the syringe (model 1002LTN; 2.5 mL; needle size: 22 ga [blunt tip]; needle length: 51 mm; Hamilton^®^, Reno, Nevada, NV, USA). A syringe-pump set to inject solution at 0.33 mL/hr was fitted to hold the cell and syringe, allowing for direct on-line measurement of the contents via ESI-MS (see Appendix A). The design of the cell allows for a designated volume (i.e., >2.5 mL) of the contents of the syringe to be continually irradiated in chorus for a set time (i.e., 15 min). The cell is adapted to permit greater volumes of solution (s), and thus alternate syringes, to be continuously irradiated. A fitted shield has also been added to protect the contents of the syringe from latent photodegradation resulting from that of standard laboratory lighting.

The stability of all RF solutions involved were also monitored via static solution-phase UV–visible absorption spectroscopy over a 4 h window (See Appendix A). Similarly, as depicted in Appendix A, static UV–visible absorption spectra of the irradiated aqueous RF samples (with/without NH_3_, respectfully) were taken by manually transferring the irradiated solution (s) from the syringe in which the solution (s) were irradiated in (model 1002LTN; 2.5 mL; Hamilton^®^) into a far-UV quartz (macro)cuvette (10 mm pathlength; Hellma^®^ UK Ltd., Essex, UK) for measurement via a conventional spectrophotometer (GENESYS 180; Thermo Fisher Scientific, Waltham, MA, USA). Notably, deionised H_2_O was used as the baseline solvent, with solutions being irradiated in 1 min intervals up to 20 min.

To differentiate any photoproducts from that of any consequential potential thermal (i.e., heat) products generated from the array of LEDs, an opaque black cloth was wrapped around the fitted syringe in a separate set of studies (see Appendix A), to remove the influence of the UVA light from the LEDs, with a temperature probe fitted to review the temperature of the experiment over time (Appendix A).

To mirror the deprotonated solutions used for gas-phase LIMS, on-line solution-phase UV photodissociation was primarily carried out on aqueous RF with NH_3_ (0.4%). However, for completeness, the solution-phase experiments were comprehensively repeated with and without the influence of NH_3_. High-resolution measurements were additionally performed on a Thermo Fisher Orbitrap mass spectrometer to allow for adequate differentiation of any (photo/thermal) product ions from that of contaminants typically found in the negative ion mode. Further analysis of the high-resolution monitoring of the reaction (s) is available within Appendix A.

## Data Availability

Data supporting results can be obtained from the authors on request.

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
