# Peer review of "Photodegradation of Riboflavin under Alkaline Conditions: What Can Gas-Phase Photolysis Tell Us about What Happens in Solution?"

_molecules, 2021, doi:10.3390/molecules26196009_

Round 1

Reviewer 1 Report

The article is presented with high scientific quality and the level of innovation, relevance, and depth of analysis are exceptional. I highly recommend the publication of this article

Author Response

We were very pleased to see that this reviewer appreciated the quality and science of our manuscript.

Reviewer 2 Report

This is a descriptive manuscript devoted to the gas-phase and solution-phase photolysis of riboflavin. The main idea of the manuscript is to compare products of photodegradation of riboflavin in gas and in solution. The most important findings of the work are: 1) the solution-phase photofragmentation closely mirrors the gas-phase photochemistry.

The manuscript is written in clear, concise style, nevertheless, some important information is missing in the Introduction and Conclusion sections. The experimental design is thought out, but some parts lack more detailed description, and I have critical comments and questions concerning the obtained results:

  1. There is no clear description, what is the main direction of the research, whether it is aimed at methodology of gas-phase photolysis comparing to solution, or the main area is riboflavin experiments. Since riboflavin and its photoreactions have been studied for a long time and are currently very well studied, I propose to shift the focus of the article to methodology. Not every modern mass spectrometric laboratory can introduce a laser into a trap and conduct such experiments. Is it purchased or home-made technique?
  2. Similar comment concerns Conclusion section. There is no clear description of why you decided to do this work, it is not enough for modern papers to conduct research for the sake of research. Please, add the Conclusion paragraph to your Discussion section and insert more specificity to the text.
  3. I have not found a description anywhere in the text of why the gas-phase photolysis of riboflavin is so important for the scientific community, except “gas-phase measurements aid the understanding of the solution-phase photochemical mechanisms”, but unfortunately no mechanisms were described in this paper, just chemical equations which are not mechanisms. Please add some information on the importance of gas-phase photolysis of RF somewhere in the Introduction section and add mechanisms to the Results section or remove the incorrect phrases from Introduction.
  4. Surprisingly I have not found Supplementary materials along with the manuscript, and it was hard to understand the half of your paper at the beginning.
  5. There are no errors on your figures, but three or five repeats were mentioned in the text. Please add it on your plots or add a separate table to the SI.
  6. Have you observed the fragmentation of the isolated ion m/z 375 before the laser flash? There are no mass spectra after accumulation and before photofragmentation. Please add this mass spectrum to the SI.

Author Response

  1. There is no clear description, what is the main direction of the research, whether it is aimed at methodology of gas-phase photolysis comparing to solution, or the main area is riboflavin experiments. Since riboflavin and its photoreactions have been studied for a long time and are currently very well studied, I propose to shift the focus of the article to methodology. Not every modern mass spectrometric laboratory can introduce a laser into a trap and conduct such experiments. Is it purchased or home-made technique?

Response: First we note that the experimental apparatus is clearly described in the methods as a custom-adapted commercial instrument.  Our previous papers (such as Ref 22) give full details.

We believe that we did explain the main goal of the paper in the abstract in the sentence “The results obtained illustrate the extent to which gas-phase photolysis methods can inform our understanding of the solution-phase photochemistry.”  To further address the comments of the reviewer, however, we are happy to have added the following sentences to the last paragraph of the introduction:

“The gas-phase photodissociation methodology employed here has an advantage compared to solution-phase measurements, as it allows the species of interest to be isolated prior to interacting with the laser.  This ensures that photofragments observed are being produced by the species of interest, and not by some minor component of the solution-phase mixture. The solution environment is potentially much more complex, since direct photoproducts can undergo secondary reactions and will be subject to environmental factors, i.e., pH, solvent, and aggregation [5,16–20]. Herein, we evaluate the benefits of this two-step methodology for studying photochemistry by applying it to RF in its native solution-phase form, [RF-H]-.  We have chosen RF for this study since it’s solution photochemistry has been well studied, thus allowing us to benchmark our solution-phase on-line photolysis technique and also understand if the gas-phase measurements can shed light on the solution-phase photochemistry.”

2. Similar comment concerns Conclusion section. There is no clear description of why you decided to do this work, it is not enough for modern papers to conduct research for the sake of research. Please, add the Conclusion paragraph to your Discussion section and insert more specificity to the text.

Response: We believe this is already addressed in our conclusions, but to further clarify this we have modified the beginning of the conclusion section to more fully address the reviewer’s remark as follows:

“In this work, gas-phase laser-interfaced photodissociation mass spectrometry has been employed on the native form of riboflavin for the first time to map its intrinsic UV-visible absorption (photodepletion) spectrum and associated photodegradation products. Solution-phase photoproducts were generated via photolysis on-line with the MS instrument, and were characterized using ESI-MS and compared with the gas-phase results.  The motivation for this work was to benchmark our on-line photolysis apparatus for a system where the solution-phase photochemistry is well known (i.e. RF), and then to compare the solution-phase and gas-phase photofragmentation to understand the benefit of conducting both measurements together.

 The on-line, solution-phase photolysis results we obtained clearly match previous solution-phase experiments conducted for RF, demonstrating that our on-line photolysis apparatus can reliably be applied to probe solution-phase photoproducts. In comparing these solution-phase results to our gas-phase results, we found that whilst [RF-H]- photodegrades into deprotonated lumichrome (m/z 241) via the loss of 134 Da in both the gas and solution phase at 365 nm (3.4 eV), further ions at m/z 255, 212, and 145 were observed to be produced following gas-phase photodissociation of the isolated molecular ion. (Our HCD results revealed that the m/z 255 ion dissociates into m/z 212 and 145 at high internal energies.)”

3. I have not found a description anywhere in the text of why the gas-phase photolysis of riboflavin is so important for the scientific community, except “gas-phase measurements aid the understanding of the solution-phase photochemical mechanisms”, but unfortunately no mechanisms were described in this paper, just chemical equations which are not mechanisms. Please add some information on the importance of gas-phase photolysis of RF somewhere in the Introduction section and add mechanisms to the Results section or remove the incorrect phrases from Introduction.

Response: The reviewer makes a good point.  We have removed the word mechanisms from the 3rd to last sentence of the introduction of the original paper, and replaced it with the correct term “photoproducts” since it is the photolysis products we have worked to identify in this publication, not mechanisms.

“The benefit of this approach is that the solution-phase measurements probe a more “real-world” environment for the riboflavin vitamin (e.g., when it is in milk or within a cosmetic formulation), while the gas-phase measurements aid the understanding of the solution-phase photochemical photoproducts.”

Note: This sentence is now further from the end of the introduction due to the additional text added in response to reviewer comment 1.

4. Surprisingly I have not found Supplementary materials along with the manuscript, and it was hard to understand the half of your paper at the beginning.

Response: There was a problem during the submission process that led to the SI not being included in the material submitted.  The SI has now been made available to the reviewers.

4. There are no errors on your figures, but three or five repeats were mentioned in the text. Please add it on your plots or add a separate table to the SI.

Response: For electronic spectra (e.g. solution-phase UV-VIS) it is not generally accepted practice to add error bars.  We have previously experimented with adding error bars onto spectra such as the data shown in Figure 2 (we assume this is the data the reviewer is referring to), but it simply leads to congestion of the data presented.  Importantly, as discussed in ref 22, there is little error (on the order of 3% per data point) so that the various repeats do not change the overall profile of the spectra.  These are highly reliable as the instrument is an adapted commercial instrument so provides a highly reliable ion signal.  

5. Have you observed the fragmentation of the isolated ion m/z 375 before the laser flash? There are no mass spectra after accumulation and before photofragmentation. Please add this mass spectrum to the SI.

Response: Figure 1a is a mass spectrum acquired before photofragmentation. Fig. 7 details how the m/z 375 fragments without a laser.

Reviewer 3 Report

Review of manuscript “Photodegradation of Riboflavin under Alkaline Conditions: What can 
Gas-Phase Photolysis Tell Us About What Happens in Solution?” by Wong, Rhodes and Dessent

The manuscript “Photodegradation of Riboflavin under Alkaline Conditions: What can Gas-Phase Photolysis Tell Us About What Happens in Solution?” reports an experimental study comparing the gas-phase and solution-phase photoreactivity of the deprotonated riboflavin anion. The authors propose an innovative method to study the time-dependent photoreactivity of a molecule in a solvent and the photostability of the same molecule in the isolated gas-phase. Such a comparison is key to better understanding the role of the solvent in photochemistry, and is not easy to do. The results of this study will be of interest both in the fundamental molecular physics and mass spectrometry communities as well as in more applied fields such as biophysics. I recommend publication after the minor comments below have been taken into account.

Language:
- While the quality of the English is high throughout, there are a few minor typos that should be corrected e.g. p2para1 “limiting curbing”, p3para1 “approaching the understanding photochemistry”

Results section:
- p6para2 “The novelty of…” The ability to monitor photofragment intensity as a function of excitation wavelength is hardly novel. This phrase should be rewritten to specifically highlight the novelty of LIMS and its application to this study.
- Figure4. Panel a should have its own y-axis label.
- Figure9. The chosen colours are difficult to distinguish in Fig9 (m242/255)

Discussion section:
- One of the major questions regarding the ESI source is the structure of the ions produced: do they represent the lowest energy solution-phase conformers, gas-phase conformers, or a mixture of both. In this study, there is a clear influence of the solution upon the observed photoproducts - the major gas-phase photoproduct at m/z 255 is not observed upon photoreactivity in solution and thus there must exist energy dissipation pathways that protect RF from this reaction pathway. The question arises: to what extent can we use gas-phase photoreactivity as a prediction of solvent phase behaviour? Some discussion should be made of these issues. The authors may wish to consider including some quantum chemical calculations to help explain why [LF-H]- is not produced in solution. Is there a structural change induced by the solvent molecules?

Methods section:
- How stable is the value IntOFF throughout the experiment?
- A comparison should be made between the photon flux during photofragmentation in the gas-phase and during irradiation in the solution. What is the photon flux of a LED and the transmission rate through the borosilicate glass at the irradiation wavelength and the glass thickness?

SI:
- Figure S2. “Previous work on the vertical detachment energies (VDEs) of flavins have shown predicted VDEs of ca. 4.0 and 3.8 eV for deprotonated alloxazine and ca. 4.6 and 4.7 eV for deprotonated structures of lumichrome.“ In the results/discussion section, please comment on the derived detachment energy around 1 eV lower than the lowest literature values for flavin VDEs. What are the sources of error in the method by which ED* is derived. Also comment on the fact that, if the ED* value that has been derived here is accurate, it is by far the majority process for the photodepletion in the anion.

Author Response

We thank this reviewer for their supportive comments on the science presented. Our responses to their queries are given below:

  1. Language:
    - While the quality of the English is high throughout, there are a few minor typos that should be corrected e.g. p2para1 “limiting curbing”, p3para1 “approaching the understanding photochemistry”

Response: These have been corrected in the revised manuscript.

  1. Results section:
    - p6para2 “The novelty of…” The ability to monitor photofragment intensity as a function of excitation wavelength is hardly novel. This phrase should be rewritten to specifically highlight the novelty of LIMS and its application to this study.

Response: The sentence was badly written and we have now rewritten it as follows to make clearer our meaning:

“Our LIMS set-up allows us to monitor the photofragment production intensities at each scanned wavelength to provide further insight into the nature of the excited states.”

3. - Figure4. Panel a should have its own y-axis label.
- Figure9. The chosen colours are difficult to distinguish in Fig9 (m242/255)

Response: Both Figures have been modified as the referee required.

4. Discussion section:
- One of the major questions regarding the ESI source is the structure of the ions produced: do they represent the lowest energy solution-phase conformers, gas-phase conformers, or a mixture of both. In this study, there is a clear influence of the solution upon the observed photoproducts - the major gas-phase photoproduct at m/z 255 is not observed upon photoreactivity in solution and thus there must exist energy dissipation pathways that protect RF from this reaction pathway. The question arises: to what extent can we use gas-phase photoreactivity as a prediction of solvent phase behaviour? Some discussion should be made of these issues. The authors may wish to consider including some quantum chemical calculations to help explain why [LF-H]- is not produced in solution. Is there a structural change induced by the solvent molecules?

Response: The reviewer raises some interesting points that could be explored in future work, although the calculations suggested are in fact extensive, and wold constitute a substantial scientific publication in their own right.  It is clearly beyond the scope of the current work to embark on this at this point.

The reviewer is correct to state that there must exist energy dissipation pathways that protect RF from breaking down into the m/z 255 fragment in solution.  We have added the following sentence to highlight this point:

“Importantly, we have observed that the major gas-phase photofragment, m/z 255, deprotonated lumiflavin, is not produced to any significant effect in solution under alkaline conditions.  This reveals that an energy dissipation pathway exists in solution to protect RF from breaking down into the 255 m/z product.”

5. Methods section:
- How stable is the value IntOFF throughout the experiment?
- A comparison should be made between the photon flux during photofragmentation in the gas-phase and during irradiation in the solution. What is the photon flux of a LED and the transmission rate through the borosilicate glass at the irradiation wavelength and the glass thickness?

Response: The value of IntOFF is very stable in this experiment as the riboflavin anion sprays very well. However, this is not relevant to the data quality since we acquire the photofragments as a laseron-laseroff

It would be valuable to determine the photon flux but this would be a challenging experiment since the ESI syringes aren’t designed to be irradiated in the way we use them in the current publication, so the manufacturers do not specify the transmission rate through glass at 365 nm.  

The reviewer has also asked for the thickness of the borosilicate glass, which is  

Barrel Inner Diameter = 0.287 inches (7.29 mm); Barrel Outer Diameter = 0.407 inches (10.3 mm); giving a thickness = 3.01 mm mm? https://www.hamiltoncompany.com/laboratory-products/syringes/81416.

  1. - Figure S2. “Previous work on the vertical detachment energies (VDEs) of flavins have shown predicted VDEs of ca. 4.0 and 3.8 eV for deprotonated alloxazine and ca. 4.6 and 4.7 eV for deprotonated structures of lumichrome.“ In the results/discussion section, please comment on the derived detachment energy around 1 eV lower than the lowest literature values for flavin VDEs. What are the sources of error in the method by which ED* is derived. Also comment on the fact that, if the ED* value that has been derived here is accurate, it is by far the majority process for the photodepletion in the anion. 
  2. Response: There is currently no discussion of the VDE of flavins in the discussion section, so we have added some further comment into the SI where Figure S2 is presented. We assume that the reviewer believes that the VDE is given by the onset of electron detachment in the ED* spectrum, but previous work from our group has shown that indirect electron detachment can occur following electronic excitation of a chromophore below the VDE.  We assume that this is the case here.   The text added states:

“While electron detachment is observed here for [RF-H]- from around 3.3 eV, it may be that indirect electron detachment below the VDE is occurring for this molecule following excitation of an electronic transition.”

Round 2

Reviewer 2 Report

Thank you for your explanationas and some modifications that you have added to the text.